# Corrosion of Steel Rebars in Anoxic Environments. Part I: Electrochemical Measurements

**DOI:** 10.3390/ma14102491

**Published:** 2021-05-12

**Authors:** Elena Garcia, Julio Torres, Nuria Rebolledo, Raul Arrabal, Javier Sanchez

**Affiliations:** 1Eduardo Torroja of Construction Science Institute (IETcc-CSIC), Serrano Galvache, 4, 28033 Madrid, Spain; elenacampaspero@gmail.com (E.G.); juliotorres@ietcc.csic.es (J.T.); nuriare@ietcc.csic.es (N.R.); 2Department of Chemical Engineering and Materials, Faculty of Chemistry, Complutense University of Madrid, Plaza de las Ciencias, Ciudad Universitaria, 28040 Madrid, Spain; rarrabal@ucm.es

**Keywords:** corrosion, anoxic conditions, reinforced concrete, chloride

## Abstract

The number of reinforced concrete structures subject to anoxic conditions such as offshore platforms and geological storage facilities is growing steadily. This study explored the behaviour of embedded steel reinforcement corrosion under anoxic conditions in the presence of different chloride concentrations. Corrosion rate values were obtained by three electrochemical techniques: Linear polarization resistance, electrochemical impedance spectroscopy, and chronopotenciometry. The corrosion rate ceiling observed was 0.98 µA/cm^2^, irrespective of the chloride content in the concrete. By means of an Evans diagram, it was possible to estimate the value of the cathodic Tafel constant (*b_c_*) to be 180 mV dec^−1^, and the current limit yielded an *i_lim_* value of 0.98 µA/cm^2^. On the other hand, the corrosion potential would lie most likely in the −900 mV_Ag/AgCl_ to −1000 mV_Ag/AgCl_ range, whilst the bounds for the most probable corrosion rate were 0.61 µA/cm^2^ to 0.22 µA/cm^2^. The experiments conducted revealed clear evidence of corrosion-induced pitting that will be assessed in subsequent research.

## 1. Introduction

Corrosion is the primary cause of shortened reinforced concrete structure service life [1,2,3], prompting the need for costly maintenance and repair [4,5,6]. That problem acquires particular significance in offshore and geological storage structures, characterised by a long service life and inaccessibility that rules out any subsequent action [7,8]. Two factors that must be borne in mind in the design of such structures are low oxygen levels and the presence of chlorides in contact with the reinforcement [9,10].
Fe → Fe^2+^ + 2e^−^     (iron oxidation)(1)
O_2_ + 2H_2_O + 4e^−^ → 4OH^−^  (oxygen reduction)(2)

The basic semi-reactions taking place in embedded steel corrosion are iron oxidation and oxygen reduction, as per Equations (1) and (2):

In underground/water environments, however, given the low levels of oxygen, the cathodic reaction is more likely to involve water reduction and hydrogen formation (Equation (3)):2H_2_O+ 2e^−^ → H_2_ + 2OH^−^ (water reduction)(3)

A number of authors have reported perceptible decay in underwater/ground structures [11,12], recording electric potentials even lower than −1.0 V_Ag/AgCl_. The inference of such low values is that corrosion is governed cathodically [8,13].

Reinforced concrete corrosion research often resorts to potentiodynamic polarisation tests, which deliver corrosion rates nearly instantaneously.

The parameters most commonly yielded in polarisation tests are corrosion potential and corrosion current density. In processes involving polarisation activation only, corrosion current density can be related to the potentiostat-measured current density with the following expression, known as the Butler–Volmer equation (Equation (4)):(4)$$i=i_{a}+i_{c}=i_{corr}\cdot \left(10^{\frac{V-E_{corr}}{b_{a}}}-10^{\frac{-\left(V-E_{corr}\right)}{b_{c}}}\right)$$
where *i* (µA/cm^2^) is the net free current density, i.e., the sum of the anodic (*i_a_*) and cathodic (*i_c_*) components; *i_corr_* (µA/cm^2^) is the corrosion current density; *V* (V) is the steel electric potential; *E_corr_* (V) is the free corrosion electric potential; and *b_a_* is the anodic and *b_c_* (V/dec) the cathodic slope of the Tafel curve. Anodic current is conventionally deemed to be positive and cathodic negative.

Table 1 below lists some of the values for the anodic and cathodic Tafel slopes and anodic and cathodic exchange current densities found in the literature.

The table shows results obtained on mortar or concrete specimens [14,15,17,18,19,21,23,24,25,26] or in alkaline solution [16,20,22]. The values listed in the table are normally the result of short-term testing conducted in which the steel was immersed in a solution. Such studies failed to dispel the uncertainty around reinforcement behaviour in concrete, which may be affected by factors such as porosity and pH [14,15]. The present in-depth study of the mechanisms and reactions taking place in reinforcement aimed to determine characteristic steel deterioration in reinforced concrete structures under anoxic conditions in the presence of different chloride concentrations. An understanding of these parameters is requisite to ensuring such structures are satisfactorily designed to the necessary durability.

Many authors claim that the corrosion rate of steel under low oxygen conditions is less than 10 µm/year. This measure has been corroborated by several studies. For example, Taniguchi et al. [27] reported that the average corrosion rates of carbon steel embedded in bentonite immersed in a synthetic marine solution between one and 10 years at 80 or 50 °C under anaerobic conditions were in the range of 0.055–1.4 µm/year, with an initial corrosion rate in the range of 10–25 µm/year, and gradually decreased with time. Kaneko et al. [28] reported the corrosion rate of carbon steel at 35 °C in a calcium hydroxide solution under anaerobic conditions simulating a concrete solution showed the initial high value of 0.2 µm/year, which decreased with time to 0.02 µm/year after 900 days. In this work, corrosion measurements were carried out directly on the steel embedded in concrete, immersed in a saline solution under conditions of a low oxygen content.

## 2. Methodology

### 2.1. Samples and Test Conditions

The tests were conducted on six 10 × 10 × 10 cm^3^ cubic concrete specimens prepared with type I 42.5R cement, batched at 350 kg of cement per m^3^ of concrete. The aggregate dosage per m^3^ of concrete was AF-0/4 766.00 kg/m^3^, AG-4/12 823.60 kg/m^3^, and AG-12/20 325.60 kg/m^3^, whilst the water/cement ratio was 0.45. The mixing water used contained chlorides in the form of sodium chloride (NaCl) at concentrations ranging from 0% to 2%, referred to cement weight. Specimens E01 and E02 had no chloride in the mixing water, whilst specimens E11 and E12 bore 1 wt% Cl^-^ and E21 and E22 2 wt% of the ion. Duplicate specimens were prepared of all the materials. Table 2 summarizes the materials and tests done.

All specimens were fitted with a 6 mm B500SD steel bar. The photo in Figure 1a depicts the experimental setup, in which the concrete was immersed in a NaCl solution and connected to the electrodes. As the photo in Figure 1b shows, prior to embedment, the bars were taped at both ends and coated with epoxy resin at the un-embedded end. The area exposed to the concrete amounted to approximately 9.7 cm^2^.

After curing in a humidity chamber for 1 d, the specimens were removed from the moulds and placed in a 30 g/L NaCl solution to conduct the electrochemical tests. The entire assembly was stored in a glovebox where the oxygen was displaced by flowing nitrogen. The glovebox oxygen content was determined indirectly by measuring CO_2_.

### 2.2. Electrotechnical Techniques

#### 2.2.1. Chronoamperometry

The oxygen concentration in the glovebox atmosphere was less than 1% throughout the experiment. Such a low oxygen content was verified with chronoamperometric readings at potentials of −750 mV_Ag/AgCl_ and −850 mV_Ag/AgCl_ [16]. The results (Figure 2) showed that the oxygen content was negligible, for negative current density values are associated with oxygen reduction whilst values near zero are indicative of an inert atmosphere and those above zero of anodic behaviour in the steel.

Three techniques were deployed to assess specimen electrochemical behaviour: linear polarisation resistance (LPR); electrochemical impedance spectroscopy (EIS); and chronopotentiometry (CP). All three measure the open circuit corrosion potential (OCP) and polarisation resistance (*Rp*) values from which corrosion rate can be calculated. The EIS and CP findings can also be used to determine the resistance drop (*R_Ω_*) value. All the readings were recorded on a Metrohm Autolab PGSTAT204 potentiostat/galvanostat. In the electrochemical cell used, saturated concrete was the medium, a steel bar the working electrode, stainless steel mesh the auxiliary electrode, and saturated Ag/AgCl solution the reference electrode. The electrochemical readings were logged over 232 days on duplicate samples.

#### 2.2.2. Linear Polarisation Resistance (LPR)

A polarisation value ±20 mV relative to OCP and a scanning rate of 0.166 mV/s were applied to estimate linear polarisation resistance (*LPR*) and subsequently to calculate the corrosion rate [17]. *Rp* was taken as the slope on the electric potential—cell current (minus the resistance drop) curve (Figure 3). Corrosion rate, *i_corr_*, was found with the Stern–Geary equation [18]:(5)$$i_{corr}=\frac{B}{Rp\;A}$$
where *B* is a constant equal to ~26 mV in concrete-embedded steel [19] and *A* is the lateral exposed area.

#### 2.2.3. Electrochemical Impedance Spectroscopy (EIS)

EIS consists of applying an electric potential of varying frequency to an electrochemical cell and measuring the complex field of the resulting current. The present study applied 37 frequencies ranging from 1 kHz to 5 mHz with an amplitude of 0.010 V [20,21]. The system’s electrochemical response was simulated with a time-constant, Randles-type equivalent circuit [22] (Figure 4). Resistance drop and linear polarisation resistance were found by fitting a regression line to the empirical data.

#### 2.2.4. Chronopotentiometry (CP)

In this test, current is pulsed through the cell at an intensity of 2.0 × 10^−5^ A and the resulting potential is monitored. Here, the linear polarisation resistance was calculated assuming the response to fit the curve for the equivalent circuit shown in Figure 3. Since the variations in potential were obtained at constant intensity, the respective resistance could be found with Ohm’s law. The initial rise in potential was associated with an ohmic drop and the remainder with linear polarisation resistance (Figure 5).

#### 2.2.5. Cyclic Polarisation Curves

This potentiodynamic electrochemical technique delivers current density across a wide range of potentials. Cyclic polarization can be used to find the Tafel slope as well as steel corrosion, pitting, and passivation rates. Here, the range defined was −1.2 V to 0.8 V, scanned at a rate of 0.1667 mV/s. Cyclic polarization tests were performed on all samples under anoxic conditions at the end of the corrosion tests.

After the electrochemical tests, concrete specimens were broken to visualise the surface condition of the steel bars under an Olympus SZ261 trinocular stereomicroscope.

## 3. Results and Discussion

### 3.1. Corrosion Rate

Figure 6 graphs the corrosion current density (*i_corr_*) for specimen E21 under anoxic conditions over a total of 232 days. The values calculated from all three techniques used in the study are plotted (although some of the EIS-based values were omitted due to measuring error). As the figure shows, the findings were very similar for all three techniques and most lay within the shaded area bounded by double and half the mean corrosion rate.

The threshold corrosion current density established to distinguish between activated and passivated concrete-embedded steel is 0.2 µA/cm^2^ [23]. Periods exhibiting active (*i_corr_* > 0.2 μA/cm^2^) and passive (*i_corr_* < 0.2 μA/cm^2^) behaviour were observed to alternate in the specimens studied, irrespective of the chloride content in the mixing water.

The corrosion rate over time plots in Figure 7 for all the specimens tested show that all six exhibited both active and passive behaviour, i.e., with corrosion rates ranging across values greater and less than the 0.2 µA/cm^2^ activation threshold for concrete-embedded steel.

### 3.2. Activation/Passivation Intervals

The mean of the three corrosion rates found with the techniques described and the mean corrosion potentials (calculated from LPR) are graphed in Figure 8 for specimen E21. The figure shows that the corrosion potential can drop to very low values, even <−1000 mV_Ag/AgCl_. Moreover, during much of the test time, potentials were below −350 mV_Ag/AgCl_, the threshold for distinguishing between passivated and activated concrete-embedded steel. Under the anoxic conditions used here, however, due to the change in the cathodic reaction and H_2_ release [7,8], the specimens exhibited passive behaviour at potentials of under −350 mV_Ag/AgCl_. The experimental findings for specimen E21 are given by way of example, for all the specimens tested exhibited similar behaviour throughout the test period.

The time intervals when the specimens were active and passive graphed in Figure 9 are based on the mean for specimens with the same chloride content. Periods in which the samples remain passive are indicated in green, while periods in which the corrosion rate is higher than 0.2 µA/cm^2^ are shown without colour. Only the specimens with no chloride in the mixing water exhibited passive behaviour at the outset. Although the timing differed, given the intermittent activation and passivation observed for the three groups, their behaviour was deemed not to differ substantially.

### 3.3. Corrosion-Induced Section Loss

Section loss (*P_corr_*) assuming uniform corrosion [24,25] (Figure 10) was estimated using the following equation based on Faraday’s law:(6)$$P_{corr}=\frac{W_{Fe}}{n*F*\rho }\int I_{corr}dt$$
where *W_Fe_* is the molecular weight of Fe, *n* the number of electrons transferred, *F* the Faraday constant, and *ρ* is iron density.

The smallest corrosion-induced section loss was observed in the specimens with no chlorides in the mixing water, whilst the highest cumulative corrosion rate was found for the specimens with 2 wt% Cl^−^. In other words, the higher the chloride concentration in the mixing water, the greater the cumulative section loss. Nonetheless, as Figure 10 shows, the differences were minor and in some intervals, specimens with a lower chloride content exhibited greater section loss. The similarity of the slopes on the three curves would indicate that the ceiling or maximum corrosion rate was the same irrespective of chloride concentration under these conditions.

### 3.4. Cyclical Polarisation Curves

Polarisation curves were found for the 232-day-old specimens while still in an anoxic environment to study their liability to pitting and to determine the experimental Tafel slope values. One of the potentiodynamic curves plotted for a scanning range of −1.2 V to 0.8 V is shown in Figure 11. The analysis was conducted starting at that very low potential because in low oxygen environments, the corrosion potential is expected to lie around −1 V. Here, it was found to be −960 mV. The mean *b_c_* value for all the tests was 210 mV dec^−1^, which is consistent with the values reported in the literature [11,26,29]. A wide band (I) spanned the graph from −650 mV to −350 mV. According to the literature, that band is normally divided into two narrower peaks located around −600 mV and −400 mV, which is consistent with iron cyclic voltammetry studies [20], for they concur with Fe(OH)_2_ oxidation to Fe_3_O_4_ and Fe_3_O_4_ to Fe_2_O_3_, respectively. The passivation potential (II) observed was −250 mV.

After the electrochemical tests, concrete specimens were broken to visualise the surface. As the example in Figure 12 shows, the bars exhibited local deterioration in the form of pits (three to four on each bar) and rust stains. The analysis of the pitting geometry will be addressed in a second part, where the depth of the pitting and the relationship between the anodic and cathodic area will be established.

### 3.5. Tafel Parameters Estimation of Current Limit Density

The Evans diagram in Figure 13 graphs the corrosion rates calculated against the corrosion potentials measured. The values were observed to range across the entire spectrum of potential values. A qualitative analysis showed that the data population for samples with no chlorides in the mixing water was denser on the lower right end of the curve, i.e., exhibiting higher potentials and lower corrosion rates. Conversely, the upper left, defined by a plateau attributed to the ceiling rate, had a higher concentration of data recorded for the specimens bearing 2% chlorides in the mixing water. Those findings are consistent with the observations discussed in connection with Figure 9 and Figure 10: although the differences were narrow, the concrete specimens with the highest chloride content underwent greater corrosion and remained activated longer than those with the lowest.

Mathematical calculations were performed to fit all the data to a Tafel curve, factoring in the effect of the current limit, *i_lim_*:(7)$$i_{corr}=\frac{i_{c}}{1+\frac{i_{c}}{i_{lim}}}$$

The cathodic Tafel constant *b_c_* was calculated to be 180 mV dec^−1^, a value consistent with those reported in the literature [11,26,29] and the mean values obtained from the polarisation curve. The current limit, *i_lim_*, was 0.98 µA/cm^2^. Most of the values lay in the interval bounded by double and half the fitted curve values. Many of those that did not lie in that interval were obtained at early test ages when the concrete contained small amounts of oxygen due to specimen preparation.

Concrete resistivity measurements made in an anoxic environment in all the specimens with the Wenner method [30,31,32,33] yielded values ranging from 160 Ω·m to 210 Ω·m. The highest values were measured in specimens with no chlorides and the lowest in those with 2% chlorides in the mixing water. Further to Gulikers’s theory [34] on the relationship between concrete resistivity and the corrosion rate, under the present test conditions, corrosion was not governed by concrete resistivity.

An analysis of the empirical data, assuming all formed part of a single population, yielded the distribution shown in Figure 14, where the log of the corrosion rate is plotted on the ‘*x*’ axis, potential on the ‘*y*’ axis, and the number of measurements recorded on the ‘*z*’ axis. As the figure shows, irrespective of the chloride content in the concrete, the potential was most likely to lie in the −900 mV_Ag/AgCl_ to −1000 mV_Ag/AgCl_ range. The bounds for the most probable corrosion rate, in turn, were 0.61 µA/cm^2^ to 0.22 µA/cm^2^ or equivalently, 7.0 µm/year to 2.6 µm/year. Normalising the values in Figure 10, which are measured to 232 days, gave corrosion depths of 5.0 µm/year to 2.0 µm/year, which are very similar to the characteristic values obtained by distributing the data.

## 4. Conclusions

The similarity among the *Rp* values obtained with the three techniques denoted high reproducibility. Under anoxic conditions, the potentials measured were less than −350 mV_Ag/AgCl_ practically throughout the test, even when the corrosion rate exhibited presumably passive values.

Under the working conditions used here, the steel was not permanently passivated. The corrosion rate, however, was independent of the chlorides present in the concrete. There are small differences that are not significant to distinguish this effect.

Graphed on an Evans-type diagram, the experimental findings used to calculate the current limit yielded an *i_lim_* value of 0.98 µA/cm^2^. At 180 mV dec^−1^, the cathodic Tafel constant (*b_c_*) found is consistent with the values reported in the literature and the mean values obtained from the polarisation curve.

The mathematical formula applied to fit the data established the theoretical values for estimating corrosion rate, affording a fuller understanding of structural durability in anoxic environments. The same formula showed that irrespective of the concrete chloride content, the potential would lie most likely in the −900 mV_Ag/AgCl_ to −1000 mV_Ag/AgCl_ range, whilst the bounds for the most probable corrosion rate were 0.61 µA/cm^2^ to 0.22 µA/cm^2^.

Although corrosion was assumed here to be uniform for the intents and purposes of corrosion depth calculations, the experiments conducted revealed clear evidence of corrosion-induced pitting that will be assessed in subsequent research.

## Figures and Tables

**Figure 1 materials-14-02491-f001:**
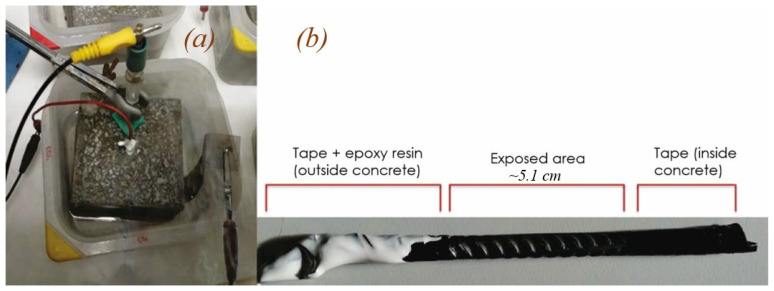
(**a**) Experimental setup; (**b**) detail of bar prepared for testing and area to be exposed to concrete when embedded.

**Figure 2 materials-14-02491-f002:**
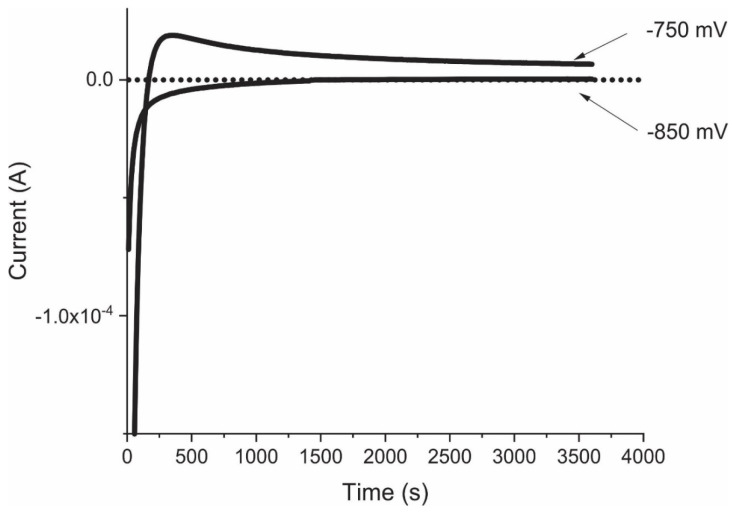
Chronoamperometry readings at −750 mV_Ag/AgCl_ and −850 mV_Ag/AgCl_.

**Figure 3 materials-14-02491-f003:**
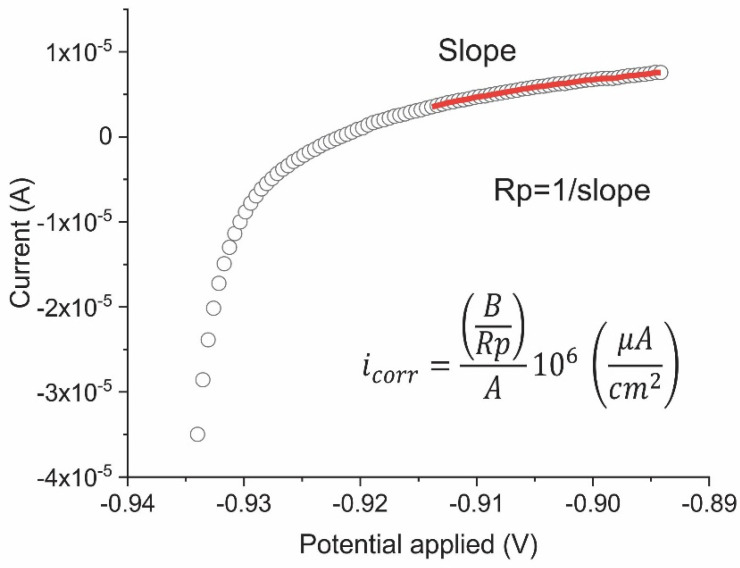
Example of the LPR test and Rp determination.

**Figure 4 materials-14-02491-f004:**
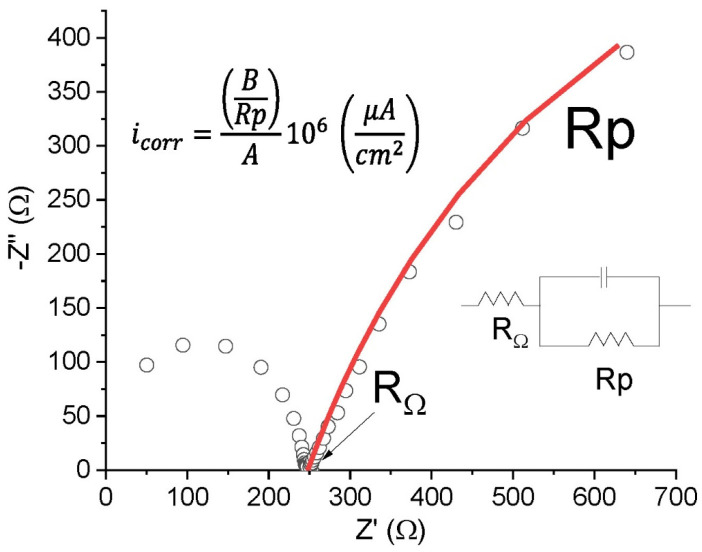
Example of EIS test and determination of R_Ω_ and Rp.

**Figure 5 materials-14-02491-f005:**
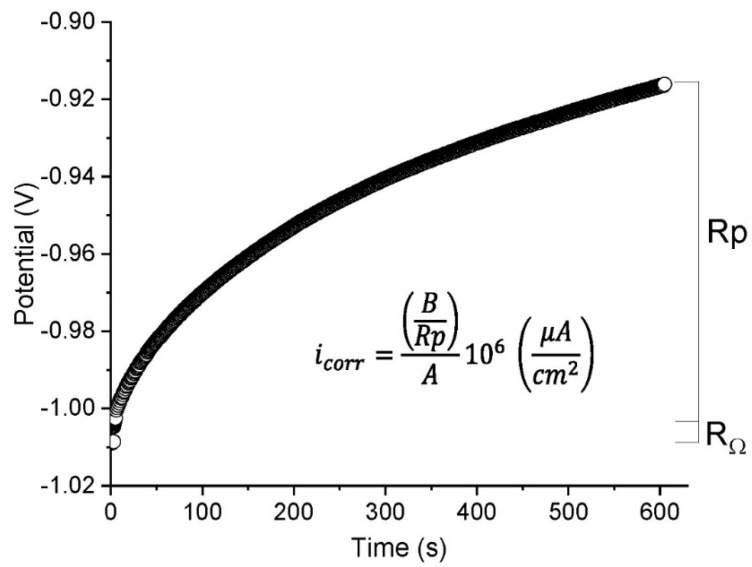
Example of the chronopotentiometry test and determination of R_Ω_ and Rp.

**Figure 6 materials-14-02491-f006:**
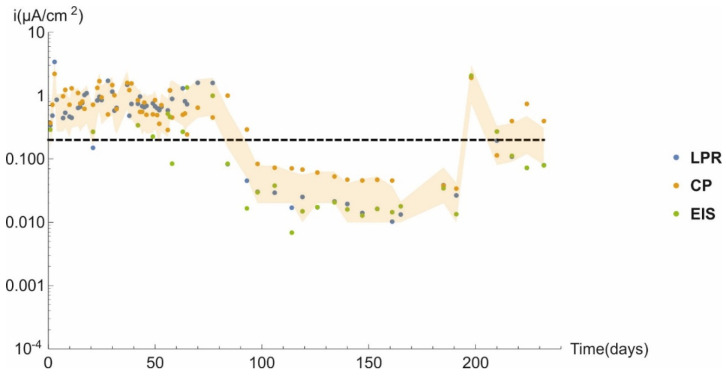
Corrosion rate over time in specimen E21.

**Figure 7 materials-14-02491-f007:**
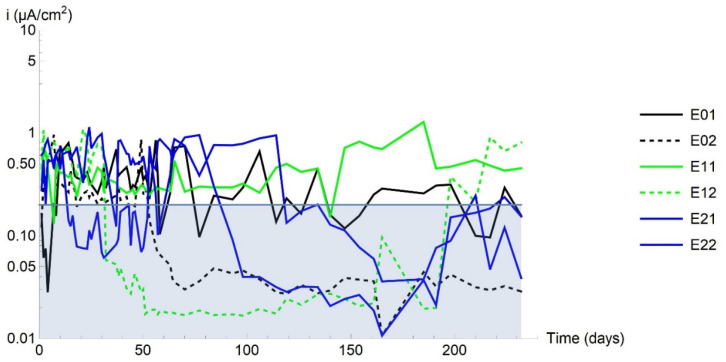
Corrosion rate over time in all the tested specimens.

**Figure 8 materials-14-02491-f008:**
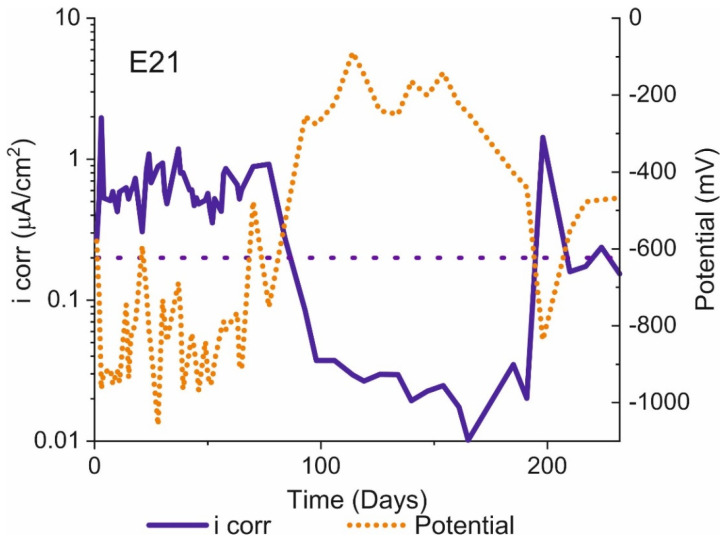
Specimen E21: i_corr_ and E_corr_ over time.

**Figure 9 materials-14-02491-f009:**
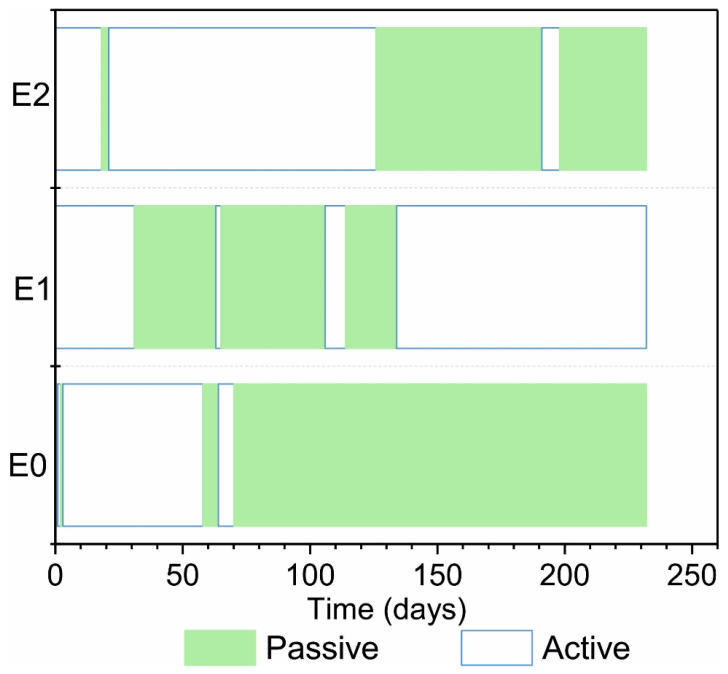
Activation and passivation timing, assuming a threshold value of 0.2 µA/cm^2^.

**Figure 10 materials-14-02491-f010:**
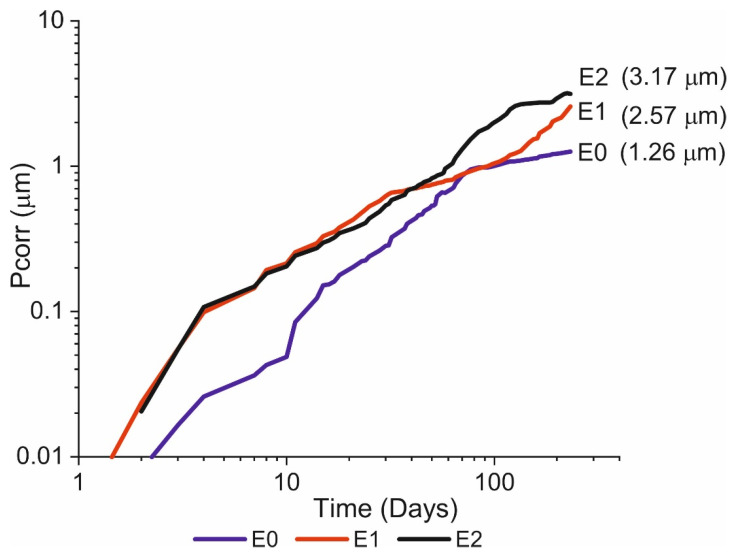
Cumulative section loss in specimens E0, E1, and E2, assuming uniform corrosion.

**Figure 11 materials-14-02491-f011:**
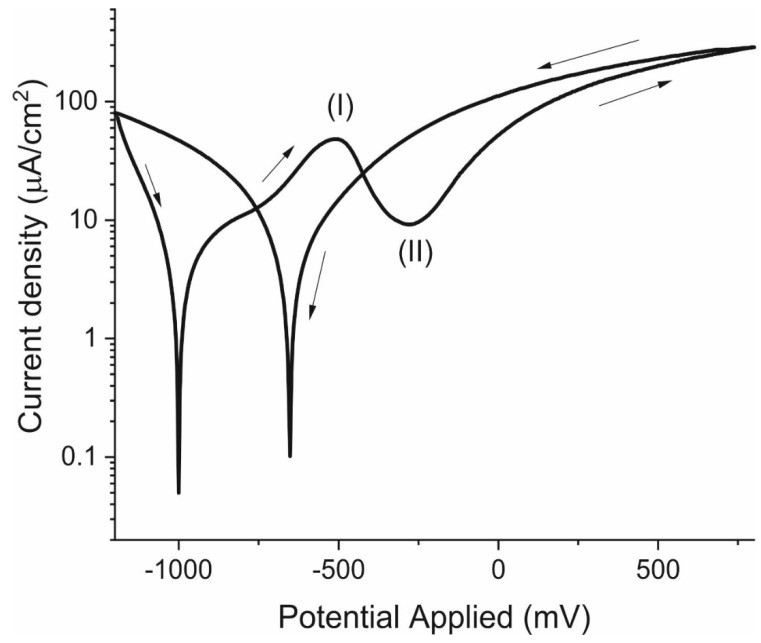
Passivation curve for specimen E21 after 232 days under anoxic conditions.

**Figure 12 materials-14-02491-f012:**
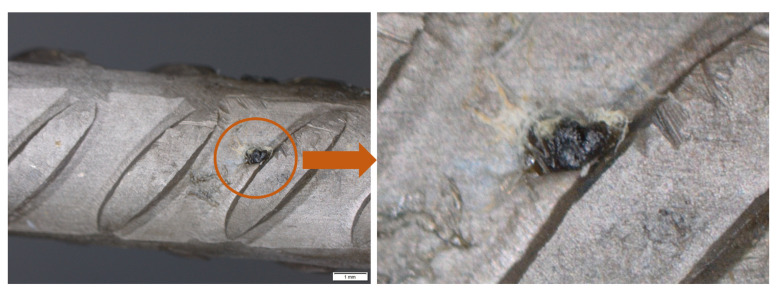
Detail of corrosion in the bar of specimen E11: overview (**left**) and close-up of a pit (**right**).

**Figure 13 materials-14-02491-f013:**
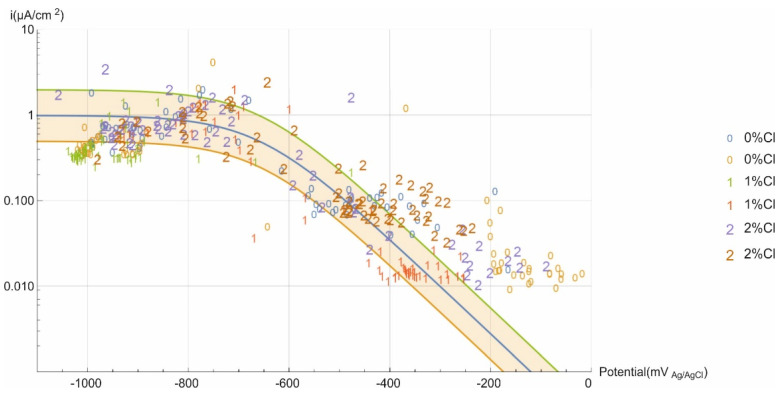
Evans diagram for the experimental data.

**Figure 14 materials-14-02491-f014:**
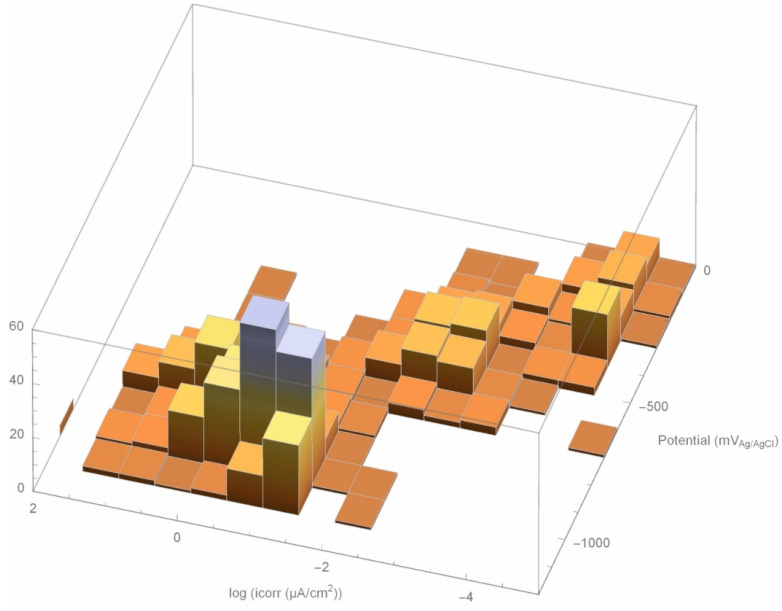
Population diagram based on the Evans diagram.

**Table 1 materials-14-02491-t001:** Summary of Tafel slopes and current densities found in the literature.

Parameter	Value	Unit	Reference
Anodic Tafel slope	0.073–0.136	V/dec	[14]
	0.067–0.080	V/dec	[15]
	0.013–0.017	V/dec	[16]
	0.320–1.570	V/dec	[17]
	0.077–0.089	V/dec	[18]
	0.01–0.369	V/dec	[19]
	0.09	V/dec	[20]
	0.075	V/dec	[21]
	1.00 × 10^−12^	V/dec	[22]
Cathodic Tafel slope	0.112–0.242	V/dec	[14]
	0.222–0.239	V/dec	[15]
	0.021–0.065	V/dec	[16]
	0.380–1.080	V/dec	[17]
	0.449–1.215	V/dec	[18]
	0.01–0.233	V/dec	[19]
	0.176	V/dec	[20]
	0.3–100	V/dec	[21]
	0.2	V/dec	[22]
Anodic exchange current density	1.00 × 10^−12^–2.75 × 10^−10^	A/mm^2^	[7,23]
	3.07 × 10^−8^–1.46 × 10^−7^	A/mm^2^	[15]
	1.73 × 10^−8^–1.69 × 10^−7^	A/mm^2^	[15]
	1.00 × 10^−12^–1.00 × 10^−9^	A/mm^2^	[19]
	1.00 × 10^−12^	A/mm^2^	[20]
	3.75 × 10^−10^	A/mm^2^	[24]
	1.88 × 10^−10^	A/mm^2^	[7]
	1.00 × 10^−12^	A/mm^2^	[24]
Cathodic exchange current density	6.00 × 10^−12^–1.00 × 10^−11^	A/mm^2^	[7,23]
	6.00 × 10^−11^–1.60 × 10^−10^	A/mm^2^	[15]
	1.00 × 10^−12^–8.50 × 10^−10^	A/mm^2^	[15]
	1.00 × 10^−12^–1.00 × 10^−9^	A/mm^2^	[19]
	1.00 x 10^−13^	A/mm^2^	[20]
	1.25 x 10^−11^	A/mm^2^	[24,25]
	6.25 × 10^−12^	A/mm^2^	[7]
	1.00 × 10^−10^	A/mm^2^	[24]
	1.00 × 10^−10^	A/mm^2^	[26]

**Table 2 materials-14-02491-t002:** Summary of dosage, samples, and tests conducted.

**Dosage**	**Material**		**Quantity (kg/m^3^)**
	Cement		350
	Aggregate	AF-0/4	766.0
		AG-4/12	823.6
		AG-12/20	325.6
	water/cement ratio		0.45
**Chloride content**	**Specimen**		**% cem**
	E01–E02		0
	E11–E12		1
	E21–E22		2
**Electrochemical techniques**	Chronoamperometry		Initial test
	Linear polarization resistance Electrochemical impedance spectroscopy Chronopotentiometry		During 232 days every 2–4 days
	Cyclic polarisation curves		Final test

## Data Availability

Data available in a publicly accessible repository.

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
