# Peer review of "Corrosion of Steel Rebars in Anoxic Environments. Part I: Electrochemical Measurements"

_materials, 2021, doi:10.3390/ma14102491_

Round 1

Reviewer 1 Report

In this manuscript the authors explored the behaviour of embedded steel reinforcement corrosion in anoxic conditions in the presence of different chloride concentrations by using a three electrochemical techniques. They stated that the corrosion process is governed cathodically and limited by hydrogen generation and the corrosion current ceiling was similar, irrespective of the chloride content in the concrete.

The presented manuscript of the authors are interesting. The some issue are not clearly described in manuscript. Therefore before publishing should be consider the following comments:

  1. Abstract is too general. Please see the journal's guidelines.
  2. In introduction should be not introduced a basic reactions about steel corrosion(Equation 1, 2, 3). In my opinion those reaction are too basic in scientific paper.
  3. In the introduction, Table 1 shows literature data obtained in other studies. The corrosion test conditions should also be provided for each publication.
  4. In chapter "Cyclic polarisation curves" the authors define the corrosion potential range and scanning rate. What was the time of measurement and whether one or several samples were tested? In this potentiodynamic test also were tested a six samples?
  5. I propose present the tested samples, the methods and corrosion parameters used in the table (in chapter 2.1).
  6. In Figure 7, the colors of the E02, E12 and E22 curves are very similar. It is difficult to accurately track changes on the chart.
  7. Please comment on Figure 9 in more detail.
  8. In chapter "Corrosion-induced section loss" please refer to literature reports.. What did other authors get in their research?
  9. Figure 12 present a detail of corrosion in bar E11 but is too small magnification. Besides in the right picture lack of bar. In my opinion the corrosion pits should be observed at much higher magnifications andit would be worth doing EDS chemical composition study if possible.
  10. Conclusions should be modified taking into account the above comments.

Reviewer 2 Report

The manuscript entitled “Corrosion of steel rebars in anoxic environments. Part I: electrochemical measurements” explored the behavior of embedded steel reinforcement corrosion in anoxic conditions in the presence of different chloride

concentrations.

The experimental work should interest readers. However, the presentation should be improved based on the following technical comments before publication.

Technical Comments:

  • The manuscript could benefit greatly from professional editing to improve technical writing and English.
  • The authors should increase their discussion on previous related research and highlight how their study is providing a different approach or adding significantly to what has been done.
  • The authors present a short abstract. The methodology used to investigate this behavior should be highlighted in the abstract by mentioning the used experimental technique.
  • Section 2.1: This reviewer highly recommends setting up a table showing more details about the tested specimens like name, components, and the number of samples.
  • Section 2.1: It is recommended to include a table to show the dosages of the concrete ingredients. Also, what do you mean by aggregates: AF-0, AG-4/12, and AG-12/20?
  • Lines 84-85: This specimen is unclear to this reviewer. Do you mean you prepared 12 specimens?
  • The embedded length of the steel bar should be illustrated in Figure 1. I think the area exposed to the concrete is not enough.
  • Line 89: What type of epoxy did the authors use? Also, what is the function of using this epoxy? Is it for increasing the bond between the concrete and steel bar surface?
  • No need to mention right or left. The Figure should be labeled as a and b. These labels should be used in the text.
  • Line 125: The term "icorr" should be written beside "Corrosion rate".
  • Also, equation 5 does not match with that equation in Figure 3. Why? This point should be clarified. I think equation 5 gives the corrosion rate for a unit area. Is that correct?
  • Line 129: What do you mean by question (A)? I cannot figure it out.
  • Figure 4: The caption of this figure does not match the term Rw. This should be fixed. The same note is for Figure 5.
  • The caption of Figure 7: I think "the tested specimens" will be more appropriate.
  • Line 227: What do you mean by "the 232 d specimens"? Do you mean for specimens at age 232 days?
  • Section 3.4: Did the authors find any change on the concrete surface around the steel bars?
  • Figure 12: The term "in bar E11" should be "in the bar of specimen E11". Also, this reviewer finds that no need to put two pictures for the same meaning. I cannot find any difference between both. It will be better to add a microscopic photo rather than a close-up one.
  • Line 296: It should be section 4, not section 5.

Round 2

Reviewer 1 Report

The authors presented responses to my comments although not exhaustive.  I think it can be published.